# The Coronavirus Pandemic and the Occurrence of Psychosomatic Symptoms: Are They Related? [note 1]

**DOI:** 10.3390/ijerph18073570

**Published:** 2021-03-30

**Authors:** Radka Zidkova, Klara Malinakova, Jitse P. van Dijk, Peter Tavel

**Affiliations:** 1Olomouc University Social Health Institute, Palacky University Olomouc, 771 11 Olomouc, Czech Republic; klara.malinakova@oushi.upol.cz (K.M.); j.p.van.dijk@umcg.nl (J.P.v.D.); peter.tavel@oushi.upol.cz (P.T.); 2Department of Community and Occupational Medicine, University Medical Center Groningen, University of Groningen, 9713 AV Groningen, The Netherlands; 3Graduate School Kosice Institute for Society and Health, P.J. Safarik University, 040 11 Kosice, Slovakia

**Keywords:** coronavirus pandemic, COVID-19, psychosomatic symptoms, health complaints

## Abstract

Most studies on the coronavirus pandemic focus on clinical aspects of the COVID-19 disease. However, less attention is paid to other health aspects of the pandemic. The aim of this study was to assess the relationship between the coronavirus pandemic (risk of infection by virus together with associated measures taken to combat it), and the occurrence of a wide range of psychosomatic symptoms and to explore if there is any factor that plays a role in this association. We collected data from a sample of Czech adults (*n* = 1431) and measured the occurrence of nine health complaints, respondents’ experience during the pandemic and sociodemographic characteristics. The results showed associations between the coronavirus pandemic and increased psychosomatic symptoms and negative emotions. We further found higher risks of increased health complaints in younger people and women. It is also possible that there is higher risk of increased health complaints for respondents with secondary school education, students, and highly spiritual people, but this relationship has to be further investigated. In contrast, respondents with their highest achieved education level being secondary school graduation had a lower risk of increased frequency of stomach-ache. We also found that more negative emotions could increase the frequency of health complaints. Our findings suggest that the coronavirus pandemic and associated government measures could have a significant influence on the prevalence of health complaints and emotional state.

## 1. Introduction

In December 2019 Wuhan, China, became the center of an outbreak of a pneumonia of unknown cause, which raised intense attention not only within China but all over the world, ultimately becoming a global threat [1]. The World Health Organization (WHO) officially named this disease coronavirus disease 2019 (COVID-19) and labelled the virus as “severe acute respiratory syndrome coronavirus 2” (SARS-CoV-2) [2]. While the vast majority of cases resulted in mild symptoms or were even asymptomatic, some cases progressed to viral pneumonia and multi-organ failure. The case fatality rate initially was estimated to range from 2–3% [3].

Depending on the expansion of the virus around the world, most state authorities of individual countries put into place various measures to stop the spread of the SARS-CoV-2 virus. For example, in the Czech Republic a state of emergency was declared (12/3/2020) and state borders were closed, as were schools, most nursery schools and most retail stores (except e.g., food and pharmacy). Free movement of people and contact with others was restricted, and only two people could stay together in public spaces at a single time (except for members of the same household). In addition, any movement or staying in all places other than the place of residence without respiratory protective equipment was prohibited [4], but at the same time there was a deficiency of masks and protective equipment. This difficult situation influenced various aspects of human life and affected the mental and physical health of individuals.

Current research has revealed a profound and wide range of psychosocial impacts on people at the individual, community and international levels during the coronavirus pandemic. On an individual level, people experienced anxiety, depression [5,6,7,8,9] and stress [6,7,8,9]; they were worried about their family members contracting COVID-19 [6], afraid for no apparent reason and were easily upset, angered or panicked [5]; they also felt fear and grief [10]. In their study, Wang et al. [6] included 1210 respondents from China and found that 53.8% rated the psychological impact of the coronavirus pandemic as moderate or severe. In addition, the prevalence of post-traumatic stress symptoms (PTSS) in China’s hardest-hit areas a month after the COVID-19 outbreak was about 7%. Furthermore, a previous study performed during the SARS-CoV-2 pandemic shows that quarantined persons exhibited a higher prevalence of symptoms of posttraumatic stress disorder (PTSD) (28.9%) and depression (31.2%), and a longer duration of quarantine was associated with an increased prevalence of PTSD symptoms [11].

Stress caused by the current situation not only may lead to psychological changes, as mentioned above, but can also affect physiological (somatic) function, leading to an occurrence of psychosomatic symptoms. A large body of research has shown significant associations between perceived stress and psychosomatic complaints [12,13,14]. The coronavirus pandemic and associated measures taken to combat it could cause people to experience high levels of stress, which can affect the prevalence of individual psychosomatic symptoms.

At present, only a few studies exist that mention the impact of the COVID-19 pandemic on the incidence of psychosomatic symptoms. Wang et al. [5] found that 6.7% of people felt tired for no reason, 12.2% had poor sleep and 6.5% were constipated. In the following study, Wang et al. [6] stated that the greater psychological impact of the coronavirus epidemic and higher levels of stress, anxiety and depression were significantly associated with poor self-rated health. Moreover, levels of anxiety were significantly associated with levels of stress, which negatively impacted quality of sleep [15]. The pandemic had an especially huge impact on the sleep of frontline medical staff, who were in close contact with patients with COVID-19. Among them, 61.7% suffered from moderate insomnia and 26.7% from severe insomnia [16].

The number of papers dealing with the impact of the coronavirus pandemic on mental and physical health is gradually increasing. Initial studies are from China (the center of the COVID-19 disease outbreak). Wang et al. [6] focused their study on psychological impact, mental health status (e.g., stress, depression and anxiety) and to identifying risk and protective factors contributing to psychological stress in the general population. Consequently, they extended their work to longitudinal study [8]. Several other studies have also addressed the psychological effects of COVID-19 [5,7,9]. Along with mental health, the effect on physical health, specifically on psychosomatic symptoms, was assessed less often and usually focused on sleep [5,6].

To the best of our knowledge, this is one of the few studies to examine the effect of the coronavirus pandemic (risk of the presence of virus together with associated measures taken to prevent the spread of the virus), on the occurrence of a wide range of psychosomatic symptoms (e.g., headache, stomach ache, backache and intestinal problems, nervousness) in COVID-19-free respondents; and furthermore, the effect is compared to the occurrence of psychosomatic symptoms before and during the coronavirus pandemic. Therefore, we want to focus our study on assessing the relationship between the coronavirus pandemic and psychosomatic symptoms that were not a direct consequence of COVID-19 and explore if there is any protective or risk factor that plays a role in this association.

## 2. Materials and Methods

### 2.1. Participants and Procedure

For this study we used data from an anonymous self-reported online survey gathered in the Czech Republic. Data was gathered during the coronavirus pandemic in April 2020 to depict the actual situation through the most critical time of the first wave of the coronavirus pandemic. In order to achieve a balanced sample regarding age and gender, a professional agency was hired to collect the data. Subsequently, to ensure high quality of the data, we have identified and excluded respondents with a unified pattern of responses and/or extremely short time filling in the survey (less than 10 min for a survey that should last around 45 min).

Next, seven respondents indicated that they had been diagnosed with COVID-19. As we wanted to evaluate the occurrence of psychosomatic symptoms that are not related to COVID-19 itself, we excluded these seven participants. Thus, the final sample contained 1431 respondents aged 18 to 97 years (mean age = 48.15; SD = 16.43; median = 47.00, Q1 = 35, Q3 = 63; CI 95%–47.30–49.90), 50.6% male.

At the beginning of the survey, participants received written information on the aim of the study and the anonymized and confidential handling of data and were made familiar with the system. In order to ensure the anonymity of respondents and given the nature of the study (an online survey gathered through a professional agency), it was not possible to use a written informed consent. Thus, an electronic one was used instead. Specifically, before starting the survey, respondents were made familiar with the content of the survey, their rights and the handling of the data and they had to explicitly express their agreement with each of the key points of informed consent. Next, they had to declare their willingness to participate in the survey by clicking on the appropriate button.

Participation in the survey was fully voluntary, so the respondents could stop responding at any time before or during the survey. The study design was approved by the Ethics Committee of the Faculty of Theology, Palacký University in Olomouc (No. 2020/06).

### 2.2. Measures

Health complaints were measured using questions focused on subjective health assessment inspired by the Health Behaviour in School-aged Children (HBSC) symptom checklist. These questions explore the prevalence of nine symptoms: (1) headache, (2) stomach-ache, (3) backache, (4) intestinal problems, (5) feeling low, (6) irritability, (7) nervousness, (8) sleeping difficulties and (9) dizziness. For each item, the respondents were asked two questions: (1) “How often did you experience the problem in the last months?”—this question examined the current situation during the pandemic, and (2) “How often did you experience the problem in a usual month before the epidemic”—to determine normal state before the pandemic. Participants answered on a five-point scale ranging from “rarely or never” (1) to “almost every day” (5). For the purposes of further analysis, any increase in incidence (e.g., from 1 to 2 as well as from 1 to 5) was considered an increase in the prevalence of health complaints, and any decrease in incidence (e.g., from 5 to 1 as well as from 2 to 1) was considered a decrease in the prevalence of health complaints.

Religiosity was assessed by the question: “At present, would you call yourself a believer?” with possible answers: (1) “Yes, I am a member of a church or religious society”, (2) “Yes, but I am not a member of a church or religious society”, (3) “No”, (4) “No, I am a convinced atheist.” Following dichotomization, respondents who answered yes (answers 1 and 2) were considered as religious and the rest (answers 3 and 4) as non-religious.

Spirituality was assessed using the Daily Spiritual Experience Scale (DSES) [17], which measures the frequency of ordinary experiences in connection with transcendence in everyday life. The scale has already been used in social psychology research [18] as well as in trauma research [19]. In the present study, an adapted 15-item version of the scale validated for the Czech environment [20] was used. Each item was evaluated on a six-degree Likert scale graded according to the intensity of experiencing the observed phenomena, ranging from “never” (1) to “many times a day” (6). A higher score corresponds to higher levels of spiritual experience. For the analysis we used the total score as a continuous variable. Cronbach’s alpha was 0.96.

Experiences during the pandemic were assessed by the question: “Has anything changed in your life in connection with the pandemic in the following areas?” These areas were: the feeling of (1) loneliness, (2) threat, (3) fear and anxiety, (4) helplessness, (5) hope. The possible answers were: (1) “worsened”; (2) “unchanged”; (3) “improved” and (4) “the question does not concern me” (this question was included because of other areas listed under the introductory question which were not connected with this manuscript). For the purpose of further analysis, the answers for each item were dichotomized such that respondents who answered 1 (worsened) were classified as experiencing an increase of negative feelings, and the rest (answers 2–4) as others.

### 2.3. Statistical Analyses

First, we performed descriptive analyses of the study sample. Further, we evaluated the change in the frequency (increase, decrease and no effect) of the occurrence of individual psychosomatic symptoms before and during the coronavirus pandemic. Due to the fact that our dependent variable was ordinal, we used the non-parametric ANOVA (Kruskal-Wallis test) for group comparison. We also evaluated the change in emotional state during the pandemic. Subsequently, we assessed if the increased frequencies of health complaints (dependent variables: headache, stomach-ache, backache, intestinal problem, feeling low, irritability, nervousness, sleep difficulties and dizziness) were related to any sociodemographic variables (independent variables: age, gender, highest achieved education, economic status, marital status, faith, spirituality). We used the Shapiro–Wilk test to evaluate the normal distribution of data. Because of the non-normal distribution of data, we assessed the associations between the increased frequency of nine kinds of health complaints and sociodemographic characteristics (age, gender, education, economic status, faith and spirituality) using binary logistic regression models, both crude (see Appendix A) and adjusted for age, gender and education. Using the binary logistic regression approach avoids the normality assumption and provides easier interpretation of the results. A Bonferroni correction was performed to correct for multiple testing. In the same way we assessed the associations of the increased frequency of nine kinds of health complaints (dependent variables: headache, stomach-ache, backache, intestinal problem, feeling low, irritability, nervousness, sleep difficulties and dizziness) with worsened feelings (independent variables: loneliness, threat, fear and anxiety, helpless and loss of hope). We performed all analyses using the statistical software package IBM SPSS version 21 (IBM, New York, NY, USA).

## 3. Results

### 3.1. Description of Population

The background characteristics of the sample are presented in Table 1. The final sample consisted of 1431 participants. Most of the respondents had finished secondary vocational school, were employees, married or in a partnership and were non-religious. Mean level of spirituality was 27.9 and median 22.0 (Q1 = 17, Q3 = 32).

### 3.2. Change in Incidence of Health Complaints during the Coronavirus Pandemic

Table 2 shows the change in the incidence of nine types of health complaints during the coronavirus pandemic compared to their incidence before it. In most respondents (ranging from 74.2% for nervousness to 94.8% for dizziness) the pandemic did not affect the incidence of health complaints. In the remaining respondents either an increase or a decrease in the prevalence of health complaints was observed; however, for all nine types of health complaints, the increase was always higher than the decrease. Increased health complaints were observed in 3.2% (for dizziness) to 17.3% of respondents (for headaches). Decreased health problems were observed in 1.7% (for dizziness) to 7.1% of respondents (for nervousness).

### 3.3. Change in Emotional State during the Coronavirus Pandemic

Table 3 shows the change in emotional state during the coronavirus pandemic compared to the time before it. In most respondents (73.9% for fear and anxiety to 90.1% for loss of hope) the coronavirus pandemic did not worsen their studied feelings. In the remaining respondents worsened feelings were observed in the range from 9.9% of respondents (for loss of hope) to 33.3% (for threat).

### 3.4. Associations of Increased Frequencies of Health Complaints with Sociodemographic Factors

Table 4 shows the associations of increased frequencies of health complaints with sociodemographic factors. Increasing age was significantly associated with a lower risk of increase of health complaints in six of the nine observed variables, with associations ranging from a 1–2% decrease in the odds ratio (*p* ˂ 0.01 and *p* ˂ 0.001) for each year. Compared to men, women were more likely to report significant increased frequencies of health complaints in three of the nine observed variables, with associations ranging from OR = 1.52 (*p* ˂ 0.01) for nervousness to OR = 2.24 (*p* ˂ 0.001) for feeling low. Regarding education, compared to respondents with elementary school, respondents with secondary school with graduation had lower chances of increased frequencies of stomach-ache (OR = 0.41; *p* ˂ 0.01). Moreover, before an application of Bonferroni correction, also respondents with secondary vocational school or secondary school with graduation were more likely to report an increased frequency of headache. Socioeconomic status did not, after an application of Bonferroni correction, have a statistically significant effect on increased frequencies of health complaints; however a tendency towards increased frequency of feeling low was observed among students. Furthermore, neither Marital status nor Faith were significantly associated with increased frequencies of health complaints, and we observed only one significant association with spirituality. Highly spiritual people were more likely to report increased frequency of dizziness, with a 2% increase in the odds ratio (OR 1.03; *p* ˂ 0.01) for each point of the DSES scale. Other associations were not significant.

### 3.5. Associations of Increased Frequencies of Health Complaints with Worsened Feelings

Table 5 shows the associations of increased frequencies of health complaints with worsened emotions. The table reveals that respondents’ psychological deterioration is associated with an increased risk of health complaints, with associations ranging from OR = 1.61 (*p* ˂ 0.01) to OR = 3.69 (*p* ˂ 0.001). The greatest effect was observed in the case of worsening feelings of fear and anxiety. These were associated with higher chances of increased frequencies of health complaints in six of the nine observed variables, with associations ranging from OR = 1.61 (*p* ˂ 0.01) for headache to OR = 3.56 (*p* ˂ 0.001) for feeling low.

## 4. Discussion

The aim of this article was to assess the relationship between the coronavirus pandemic and psychosomatic symptoms that were not a direct consequence of COVID-19 and to explore potential protective or risk factors in these associations. The results showed a significant increase of frequencies of psychosomatic symptoms and negative emotions. This involved especially younger people, women and more spiritual people. However, potentially it could also concern respondents with secondary school education and students, where we observed an increase in the frequency of psychosomatic symptoms, though this increase was not significant after Bonferroni correction. In contrast, respondents with the highest achieved education being secondary school with graduation compared to respondents with basic school had a lower risk of increased frequency of stomach-ache. No association was observed for marital status and faith. We also found that worsened emotions associated with experiencing the coronavirus pandemic could increase the frequencies of mental and physical health complaints.

We first found that, though the majority of respondents reported no change in the frequency of health problems during the coronavirus pandemic, the remainder reported rather an increase than a decrease in the prevalence of health complaints. These findings are consistent with other studies which suggested that the coronavirus pandemic is affecting both the physical [6,9,21] and mental health [6,7,8,9,10] of the population. The decrease in the frequency of health complaints could be caused by government measures, due to which some people had to stay at home and thus had more time for themselves.

Our second observation was a worsening of feelings among respondents: from 9.9% (for loss of hope) to 33.3% (for threat). These results are in line with other studies which argue that because of the coronavirus pandemic, people reported depressive symptoms, anxiety, stress [6,7,8,9] and worries about their family members [6], and they feel fear [9,10], anger [9], grief, and traumatic reactions [10]. Thus, it could be possible that during the coronavirus pandemic a wide range of emotions were affected, resulting in an overall deterioration of mental status.

We further found that in six of the nine observed health complaints, the risk of increased frequencies of health complaints decreases with growing age. These observations are in line with the findings of other authors who mention that during the coronavirus pandemic older age was associated with less psychological distress [22], less anxiety [5,23], less depression and better overall mental health [23]; older participants also reported sleep problems slightly less frequently than young people [21]. This also corresponds with a previous study suggesting that older adults tend to report fewer negative emotions, are less responsive to daily stressors and have better mental health [24]. Better mental health may also be the reason for the lower probability of an increasing frequency of physical health complaints. As research shows, there is a relation between mental health and physical health [25,26,27], where positive emotional states may promote healthy perceptions and better physical conditions [27,28]. Another explanation could be that older people naturally have more physical health problems, so their increase may not be as visible.

Our sociodemographic data also suggest that females showed a significantly higher risk of increased frequency of health complaints compared to males in three of the nine observed variables. Above that, a higher risk of increased frequencies of health complaints was observed for three other variables, which were significant before the Bonferroni correction. These findings are in line with other studies, which suggest that the coronavirus pandemic has a greater psychological effect on women. Studies show that female gender is a higher risk factor for anxiety [5,6,29,30], depressive symptoms [6,29,30], stress [6] and psychological distress [22]. Women also reported higher post-traumatic stress symptoms [31]. The greater influence on women could be caused by many factors, e.g., that women are more frequently front-line health-care workers, so they may be potentially more susceptible to virus infection [32]. Moreover, women usually had to care for children when schools were closed. Because of the relationship between the emotional and physical conditions of an individual [27], psychological problems may also lead to a change in physical health. This is also confirmed by the study of Beck et al. [21] conducted during the coronavirus pandemic, in which the authors reported that women had more sleeping difficulties than men.

We also found that economic status did not have a significant effect on increased frequencies of health complaints. A possible exception could be represented by students, where we found before Bonferroni correction a relationship between students’ status and increased frequency of feeling low. Nevertheless, further analyses on a larger sample are needed to confirm this relationship. A greater psychological impact on students was also observed in the study of Wang et al. [6,7] and in the study of Odrizola-Gonzalez et al. [7], where the authors observed a higher level of stress, anxiety and depression. Regarding the economic shutdown, which caused an increase in unemployment, some students could have been worried about unemployment when they graduate from college in the near future [10] or worried about their educational process (e.g., extending the length of study, impossibility of practice, etc.).

We further found that faith did not affect the risk of increased frequencies of health problems. This is in contrast to the findings other studies, which reported that religiosity and spirituality have positive associations with better mental health [33,34,35] and better physical health [33,36,37]. An explanation could be the different religious background of the countries. While other studies were conducted in predominantly religious countries, our study was performed in the most secular country in the world [38]. The study of Stavrova [39] suggested that the relation of religiosity with self-rated health largely depends on the regional level of religiosity. Our earlier study also suggests that in the Czech Republic, religious attendance and religious well-being for the most part do not have any impact on adolescent health complaints [40].

We further found that all assessed negative feelings had an association with increased frequencies of health complaints. As expected, fear and anxiety had the greatest effect on the number of increased frequencies of health complaints (on the six of the nine observed variables) and loneliness had the greatest effect in odds ratios (OR 3.69); however, we also observed associations with physical health. Thus, our findings further support the idea of the association between mental and physical health [25,26,27,28], as we mentioned above.

Finally, there are several findings, there are no easy to interpret. First, education did not significantly affect the risk of increased frequencies of observed variables with the exception of respondents with secondary school with graduation, who had lower chances of increased frequency of stomach-ache. To the best of our knowledge, no one has thus far studied these association; thus, in order to be able to explain our results, more detailed analyses are needed. Furthermore, before Bonferroni correction, the results showed that respondents with a secondary education were more likely to report an increased frequency of headache, which is in contrast to other studies [41]. An explanation of these discrepancies could be that none of these studies was conducted during the coronavirus pandemic, in a non-standard and psychologically highly demanding situation. However, this relationship needs to be verified in subsequent studies on a larger sample.

Second, we also found that more spiritual people were more likely to report an increased frequency of dizziness, which is in contrast with our earlier study, where we suggested that spirituality could decrease the prevalence of health complaints in Czech adolescents [40]. These discrepancies may be caused by the respondents’ age (adolescents vs. adults) or by the measurement tool (Spiritual Well-Being Scale vs. DSES). However, due to the small number of respondents with increased frequencies of dizziness, it is very difficult to interpret the data, and before making any conclusions, it is necessary to verify these results on a larger sample.

### 4.1. Strengths and Limitations

The strength of this study is that it captures the state of health complaints of respondents at the most critical time of the first wave of the coronavirus pandemic and compares this with the self-reported state before the pandemic. To the best of our knowledge, this is one of the few studies examining the association of psychosomatic health complaints with the coronavirus pandemic to such a wide extent and examining changes in the frequency of their occurrence compared to the pre-pandemic state. Another strength is the large sample that is close to national sample characteristics regarding age and gender.

A limitation of our study may be information bias, because the data are based on self-reports of the participants. Although the sample is close to national sample characteristics, selection bias could be present because of the online method of data collection (information technology skills, etc.). Next, due to the fact, that our independent–dependent variable relationship could be distorted by some other unknown factor, confounding bias could be also present. Moreover, although we excluded respondents who stated that they had been diagnosed with COVID-19, we cannot be sure that our cohort surely included only COVID-19-free individuals, and so that our observed health complaints are only of psychosomatic origin. Another limitation could be that psychosomatic symptoms are possibly affected by COVID-19 related occupations (e. g. medical staff). In our research, we did not identify these specific occupations among the participants, so we did not adjust for this in analyses. Further, it should be mentioned that the survey was performed in the early stage of the COVID-19 pandemic, so the survey does not reflect socioeconomic changes in its later phases (e.g., income change, employment change) and their consequent impact on health complaints. Finally, our design does not allow us to come to conclusions on causality.

### 4.2. Implications

Our findings suggest that the coronavirus pandemic and its associated measures aimed to prevent the spread of the virus (quarantine, economic shutdown) could affect both physical and mental health, and therefore this should be brought to the attention of the general public. Based on sociodemographic information, we identified women, youngsters, highly spiritual people and, potentially, students and people with highest achieved secondary school education as vulnerable groups, so it is important to maximize support and provide early psychological interventions. Our research results are of a great practical significance and can be immediately applied to optimize the response to similar future situations as well as to the long-term effects of the current pandemic. Results that were significant before Bonferroni correction, and after correction ceased to be significant, should be verified on a sample with a higher statistical power. Furthermore, it would be useful to assess the pathway of individual relationships and conduct a prospective study after a period to detect any long-term effects of the coronavirus pandemic on psychosomatic health complaints.

## 5. Conclusions

Our findings suggest that in the Czech Republic the initial phase of the coronavirus pandemic and associated government measures to prevent the spread of the virus influenced the occurrence of psychosomatic health complaints. Respondents also reported worsened feelings compared to the pre-pandemic period. Among the risk factors associated with a higher risk of increased frequency of health complaints are younger age, female gender and, potentially, also student status, highest achieved secondary school education and high spirituality. Further research in this area is needed with respect to the clinical applicability of these findings, e.g., by assessing the causal pathway. Moreover, a prospective study should be conducted after a period (e.g., after a half year or one year after the pandemic) to detect long-term effects of the coronavirus pandemic on psychosomatic health complaints.

## Figures and Tables

**Table 1 ijerph-18-03570-t001:** Description of the study population.

Sociodemographic Group		Total	
		*n*	%
Gender	Male	724	50.6
	Female	707	49.4
Highest education achieved	Elementary school	122	8.5
	Secondary vocational school	651	45.5
	Secondary school with graduation	475	33.2
	College	183	12.8
Economic status	Employee	707	49.4
	Self-employed	72	5.0
	Household/Unemployed	127	8.9
	Student	77	5.4
	Disabled/Old-age pensioner	448	31.3
Marital status	Single/Divorced/Widow(er)	489	34.2
	Married/Partner relationship	942	65.8
Faith	Non-religious	947	66.2
	Religious	484	33.8

**Table 2 ijerph-18-03570-t002:** The coronavirus pandemic and the incidence of health complaints.

Health Complaints	Decrease in Prevalence of Health Complaints		No Effect		Increase in Prevalence of Health Complaints		Group Differences
	N	%	N	%	N	%	*p*-Value
Headache	64	4.5	1097	76.7	270	17.3	<0.001
Stomach-ache	43	3.0	1268	88.6	120	7.7	<0.001
Backache	69	4.8	1179	82.4	183	12.8	<0.001
Intestinal problem	52	3.6	1271	88.8	108	7.7	<0.001
Feeling low	76	5.4	1098	76.7	257	16.5	<0.001
Irritability	91	6.3	1081	75.5	259	16.6	<0.001
Nervousness	100	7.1	1062	74.2	269	17.2	<0.001
Sleeping difficulties	61	4.3	1208	84.4	162	10.3	<0.001
Dizziness	24	1.7	1357	94.8	50	3.2	<0.001

Notes: group differences were assessed using the Kruskal-Wallis test.

**Table 3 ijerph-18-03570-t003:** Change in emotional state during the coronavirus pandemic.

Feelings	Increased Negative Feelings		Not Increased Negative Feelings	
	N	%	N	%
Loneliness	254	17.7	1177	82.3
Threat	476	33.3	955	66.7
Fear and anxiety	373	26.1	1058	73.9
Helpless	349	24.4	1082	75.6
Loss of hope	142	9.9	1289	90.1

**Table 4 ijerph-18-03570-t004:** Associations of increased frequencies of health complaints with sociodemographic factors adjusted for age, gender and education (OR, 95% CI).

Variables	Headache	Stomach Ache	Backache	Intestinal Problems	Feeling Low	Irritability	Nervousness	Sleeping Difficulties	Dizziness
Age	0.98 (0.97–0.98) ***	0.98 (0.97–1.00) **	1.00 (0.99–1.01)	0.99 (0.98–1.00)	0.99 (0.98–1.00) **	0.99 (0.98–1.00) **	0.99 (0.98–1.00) **	0.99 (0.98–1.00) **	0.98 (0.96–3.83)
Gender									
Male	1	1	1	1	1	1	1	1	1
Female	1.34 (1.01–1.77) *	2.19 (1.45–3.30) ***	1.17 (0.85–1.62)	1.11 (0.74–1.68)	2.24 (1.67–3.01) ***	1.33 (1.00–1.76) *	1.52 (1.15–2.01) **	1.34 (0.95–1.88)	2.05 (1.10–3.83) *
Highest achieved education									
Elementary school	1	1	1	1	1	1	1	1	1
Secondary vocation school	2.17 (1.20–3.92) *	0.56 (0.31–1.03)	1.04 (0.58–1.87)	0.64 (0.32–1.25)	0.79 (0.47–1.33)	1.21 (0.70–2.09)	0.87 (0.52–1.44)	0.79 (0.44–1.40)	0.47 (0.19–1.14)
Secondary school with graduation	2.16 (1.19–3.92) *	0.41 (0.22–0.77) **	0.84 (0.46–1.53)	0.84 (0.43–1.63)	0.96 (0.58–1.61)	1.29 (0.75–2.23)	1.01 (0.61–1.67)	0.71 (0.40–1.28)	0.50 (0.21–1.21)
College	1.70 (0.87–3.34)	0.53 (0.26–1.09)	1.19 (0.61–2.32)	0.61 (0.27–1.40)	1.36 (0.77–2.41)	1.81 (0.99–3.31)	1.35 (0.77–2.37)	0.93 (0.48–1.81)	0.51 (0.18–1.48)
Economic status									
Household/unemployed	1	1	1	1	1	1	1	1	1
Student	1.74 (0.87–3.50)	0.75 (0.31–1.84)	1.23 (0.52–2.90)	1.28 (0.52–3.16)	2.01 (1.01–3.99) *	2.01 (0.98–4.18)	1.55 (0.78–3.05)	2.15 (0.98–4.73)	1.50 (0.41–5.52)
Disabled/Old-aged pensioner	1.48 (0.78–2.80)	0.77 (0.35–1.72)	1.39 (0.67–2.90)	0.74 (0.31–1.77)	0.92 (0.49–1.74)	1.69 (0.87–3.27)	1.44 (0.78–2.68)	1.41 (0.65–3.03)	2.89 (0.79–10.64)
Employee	1.21 (0.73–1.99)	0.62 (0.34–1.23)	1.02 (0.56–1.84)	0.62 (0.32–1.21)	0.98 (0.61–1.58)	1.54 (0.91–2.61)	0.94 (0.58–1.52)	0.93 (0.51–1.71)	1.18 (0.42–3.34)
Self-employed	1.32 (0.62–2.81)	1.37 (0.56–1.00)	1.46 (0.62–3.44)	0.82 (0.29–2.34)	1.51 (0.73–3.12)	0.99 (0.42–2.36)	0.99 (0.46–2.15)	1.00 (0.38–2.64)	2.16 (0.47–9.85)
Marital status									
Single/Divorced/Widow(er)	1	1	1	1	1	1	1	1	1
Married/Partner relationship	1.07 (0.80–1.42)	1.01 (0.67–1.50)	0.98 (0.70–1.36)	0.81 (0.54–1.22)	0.84 (0.63–1.12)	0.77 (0.58–1.02)	0.79 (0.59–1.04)	1.07 (0.75–1.52)	1.04 (0.57–1.92)
Faith									
Non-religious	1	1	1	1	1	1	1	1	1
Religious	1.26 (0.95–1.67)	1.09 (0.73–1.62)	0.93 (0.67–1.31)	1.09 (0.72–1.65)	1.09 (0.82–1.45)	0.91 (0.68–1.21)	1.16 (0.88–1.54)	1.18 (0.83–1.66)	1.52 (0.85–2.72)
Spirituality	1.00 (0.99–1.01)	1.01 (1.00–1.02)	1.00 (0–99-1.01)	1.01 (0.99–1.02)	1.01 (1.00–1.02)	1.00 (0.99–1.01)	1.01 (1.00–1.02)	1.03 (0.99–1.01)	1.03 (1.01–1.04) **

Notes: * *p* < 0.05, ** *p* < 0.01, *** *p* < 0.001; after using Bonferroni correction, only *p*-values below 0.01 are considered significant; OR—odds ratios; CI—confidence intervals.

**Table 5 ijerph-18-03570-t005:** Associations of increased frequencies of health complaints with worsened feelings adjusted for age, gender and education (OR, 95% CI).

Variables	Headache	Stomach Ache	Backache	Intestinal Problems	Feeling Low	Irritability	Nervousness	Sleeping Difficulties	Dizziness
Loneliness									
Others	1	1	1	1	1	1	1	1	1
Worsened	1.42 (1.02–1.98) *	1.82 (1.18–2.80) **	1.32 (0.89–1.95)	1.48 (0.92–2.37)	3.69 (2.70–5.04) ***	3.42 (2.50–4.67) ***	2.80 (2.05–3.82) ***	1.76 (1.20–2.58) **	1.66 (0.88–3.14)
Threat									
Others	1	1	1	1	1	1	1	1	1
Worsened	1.25 (0.94–1.65)	1.19 (0.80–1.76)	1.41 (1.02–1.95) *	1.37 (0.92–2.06)	2.35 (1.77–3.11) ***	2.04 (1.54–2.69) ***	2.53 (1.92–3.33) ***	1.97 (1.41–2.75) ***	1.27 (0.71–2.28)
Others	1	1	1	1	1	1	1	1	1
Worsened	1.61 (1.20–2.17) **	1.80 (1.21–2.69) **	1.33 (0.94–1.87)	1.53 (1.00–2.35) *	3.56 (2.66–4.76) ***	2.72 (2.04–3.63) ***	3.01 (2.26–4.00) ***	2.35 (1.66–3.26) ***	2.02 (1.12–3.63) *
Helpless									
Others	1	1	1	1	1	1	1	1	1
Worsened	1.27 (0.93–1.72)	1.39 (0.92–2.10)	1.12 (0.85–1.73)	1.40 (0.91–2.17)	2.72 (2.03–3.64) ***	2.43 (1.81–3.25) ***	2.99 (2.24–3.98) ***	2.37 (1.67–3.36) ***	2.46 (1.37–4.42) **
Others	1	1	1	1	1	1	1	1	1
Worsened	1.34 (0.89–2.03)	1.25 (0.71–2.21)	1.04 (0.62–1.75)	1.42 (0.79–2.54)	2.10 (1.42–3.11) ***	2.48 (1.69–3.64) ***	2.05 (1.39–3.02) ***	1.99 (1.26–3.14) **	1.28 (0.562.93)

Notes: * *p* < 0.05, ** *p* < 0.01, *** *p* < 0.001; after using Bonferroni correction, only *p*-values below 0.01 are considered significant; OR—odds ratios; CI—confidence intervals.

## Data Availability

The data presented in this study are available on request from the corresponding author.

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
