# Peer review of "The Coronavirus Pandemic and the Occurrence of Psychosomatic Symptoms: Are They Related?"

_ijerph, 2021, doi:10.3390/ijerph18073570_

Round 1

Reviewer 1 Report

This study provided research results on the relationship between corona virus and psychosomatic symptoms(nine health complaints) in Czech adults. Although the results of the survey were analysed on large samples of Czech adults, unfortunately there were many unexplained factors(for example, more spiritual people were more likely to report an increased frequency of dizziness).   Nevertheless, as the authors described, it is a very rare paper describing the relationship between corona virus and psychological symptoms and is also considered to be a high-quality paper that describes the overall pandemic phenomenon in the Czech Republic.   In order to improve the quality of the paper, the followings are reflected or corrected.
- Describe whether authors have solved the research ethics problem for Czech adults who participated in the survey. That is, they should state whether they have granted permission for the survey in writing or in a similar manner. This is because the survey is very personal and there is a possibility of privacy infringement.   -It should also be checked that the respondents responded correctly to their health and psychological conditions. Furthermore, it should be described in more detail whether the questionnaire that did not respond correctly was excluded from the analysis.   -Ii is required to insert 'Related works' section between 'Introduction' and 'Materials and Methods'. Authors need to check if this is the first real work that examines the effect of the corona virus pandemic on the occurrence of a wide range of psychosomatic symptoms.    -In Discussion Section, there are many statements starting with "we found..." with explanations. It is described in a mixture of what phenomena are explainable and which are not explainable. Thus, separately describing explainable and unexplained phenomena will more clearly represent the contributions and limitations of this study.   -In '5. Conclusions', title should be changed into '5. Conclusions and Further Research Work', and  some further research work issues need to be discussed.

Reviewer 2 Report

The authors evaluated the relationship between the coronavirus pandemic and psychosomatic symptoms and explored potential protective or risk factors using the online survey for 1,438 Czech adults. Considering that most studies on the coronavirus pandemic focus clinical aspects of the COVID-19 disease, this manuscript can provide valuable information about psychosomatic symptoms, which might be related to the coronavirus pandemic, to be used for early psychological interventions in similar pandemic situations in the future.  
Although the manuscript was well written with the specific purpose of the analyses, the survey design and some parts of analysis methods were not clear, which should be clearly addressed. The manuscript can be more developed with incorporation of following comments: 
1. Because this study was based on the survey, it is important to provide more detail information about the survey design such as a sampling error and confidence interval. In addition, how did authors select target population to conduct the survey? Did the authors collect personal contact information (e.g., email) to provide the online survey link? Or, did the authors provide the survey link on certain webpages (e.g., government webpage for COVID-19 announcement)? 
2. Psychosomatic symptoms are possibly affected by COVID-19 related occupations (e.g., medical staff). Would it be possible to identify these occupations among the survey participants, and adjust this for the analyses?
3. Socioeconomic status should be generally very important factors for psychosomatic symptoms or psychological deterioration. Thus, it is critical to examine the change of socioeconomic status after COVID-19 pandemic. It seems that authors used the socioeconomic status of the survey participants at the time of the survey which didn’t reflect change of the socioeconomic status after COVID-19 pandemic (e.g., income change, employment change). If it is not possible to consider this information in this study, the authors should address this limitation in the manuscript. 
4. The authors used Bonferroni correction for the multiplicity adjustment. However, it is not clear how to change the significance level from 0.05 to 0.01? Is the change based on the number of testing or the number of outcome variables? Both cases didn’t seem to be relevant to the change from 0.05 to 0.01. Otherwise, the authors change the significance level arbitrarily? Please address Bonferroni correction for the change of significance level from 0.05 to 0.01. 
5. The authors used some confounders (e.g., age, gender, education) for the analyses to examine the association between health complaints and worsened feelings. Why the authors didn’t include other variables like faith (spiritual people), which was related to dizziness, as confounders. Please address how to select confounders. For the purpose of the analysis, it wouldn’t harm to adjust all the variables as confounders. 

Reviewer 3 Report

The manuscript entitled: ”The coronavirus pandemic and the occurrence of psychosomatic symptoms: are they related?” interestingly focuses on the possible psychosomatic disorders that the still current coronavirus pandemic could cause on a Czech Republic’s population, as an indirect effect. 

Abstract: It is not clear what the Authors mean with: “measures to combact it”. They do not take into account any measure, and they do not put in correlation them with the occurrence of psychosomatic symptoms. Also at the end of the abstract (line 25), Authors reiterate:  “that the…and associate government measures…”, Authors have not analysed data which have allowed to conclude this evidence, though I agree, the lockdown has negatively affected the emotional state of people. Please rephrase here and in the final paragraph of the introduction section.

M&M: More info should be added to explain how participants were urged to participate. Where the questionnaire was advertised. The age ranged from 18- to 97-year-old, not common for an elderly to be familiar with technology. Median and IQRs are more appropriate describing non-parametric variables. All dependent and independent variables used for analysis should be listed here, this helps readers to understand the results. The reason why Authors have opted to use the Bonferroni correction, a really conservative test, should be given.  

Tables: Some error and results to correct are present. In Tables 1, 4, 5 predictors, such as gender, highest education achieved or loneliness, threat and so on, should be placed in an extra column on the left making more understandable what the variables and the categories are. In Tables 2 and 3 “Column total” and “Total” respectively, are in the wrong place. I think readers know that the respondents are 1,431, so to report the total is useless and confounding. However, in table 2 something is wrong, the backache line reports 67 who have benefit, 1,179 with no effect and 183 with negative effect, so the total is 1,429. Furthermore, table 3 shows that the most frequent emotional state reported is Threat (33.3%), conversely lines 184 and 247 state “for fear and anxiety”.  Please check.  Tables 4-5, please remove 1 and replace with baseline.

Discussion: Lines 228-229, Authors state that the results showed a significant increase of frequencies of psychosomatic symptoms and negative emotions. Data do not show a marked difference between a decrease in prevalence and increase in prevalence for health complaints, and as for feelings too. So, this sentence should be mitigated or further analysis are needed to bear the hypothesis. At the end of the discussion Authors report that “fear and anxiety” have the greatest effect, this is wrong, as table 5 shows the highest effect (OR 3.69). Please check. Authors in the Limitation paragraph list limit and bias, other biases present are selection bias and confounding bias that should also be discussed.

Overall, the manuscript offers interesting results which need to be interpretated with caution.

Reviewer 4 Report

This is an interesting study performed during the first wave of coronavirus pandemic. The main objective was assessing the association between the coronavirus pandemic and psychosomatic symptoms that were not a direct consequence of COVID-19 in a large sample (n=1,431).

Introduction

- Lines 58 and 76: is it referred to “pandemic situation” instead of “epidemic situation”?

Material and methods

- Line 99: The data about the proportion of male in the study population is repeated twice (line 99 and 164). I will recommend describe it in the only place of the text.

- It is possible to explain more data about “participants and procedure”? For example, I would like to know: how the subjects were invited to participate, whether the questionnaire was self-reported or not, how it was known whether the subjects were diagnosed with COVID...

- Line 109: Is the Health Behaviour in School-aged Children symptom checklist appropriate to use in the study population? (mean of age was 48.15 years).

- Line 112: to measure dream difficulties, why have other validated scales not been taken into account? (Athens Insomnia Scale, Pittsburgh Sleep Quality Index..)

Results

- Table 1: the name of the variables could be in bold to differentiate them from the name of the categories

- Table 3: to show the comparison between the emotional state during coronavirus pandemic and normal time before it, you can indicate a p-value.

If “others” are those participants that did not worsen their studied feelings, why are not they called for example, "Not increased negative feelings?

The name of Table 3 maybe should be more specific

- Table 4: Table 4 could be placed on a horizontal sheet for better visualization of it. The abbreviations should be in the footnotes of the table instead of in the title

- Line 219: This result (OR= 1.61 (p< 0.01) is not according to Table 5, maybe did you meant "for headache”?

- Table 5 have another type of letter. Maybe, the confidence interval could be placed in a second row. The abbreviations should be in the footnotes instead of in the title

Discussion

- Line 247: According to table 3, 33% is referred to "threat"

Supplementary Material

I can't find the difference between table 4 and the supplementary one. Material supplementary's table is not cited in the text.

Round 2

Reviewer 2 Report

The authors have adequately addressed the points raised in my previous review except Bonferroni correction.

The authors answered that the change of the significance level from 0.05 to 0.01 was based on the number of testings (5), but I didn't understand how the authors counted the number of testings used in the current manuscript. The number of dependent or indenpendent variables are more than five and thus, probably the total number of testings should be more than five.  If the authours selected the testings arbitrarily for setting significance level of 0.01, it was wrong. I suggest that authors don't use the Bonferroni correction. As the study design is not a confirmatory study (i.e., exploratory purpose), it woun't hurt not to use Bonferroni correction. 

Reviewer 3 Report

Dear Authors,

thank you for having addressed all the points requested. Just one more correction: IQRs need to be reported 25th (known as Q1) and 75th (Q2) percentile, not just one (median=47.00; IQR = 28;).

Best regards

Reviewer 4 Report

The manuscript has been improved compared to the previous version; however, the following aspects must be addressed prior to its publication:

- In the footnotes of Table 2 it was specified “confidence intervals 95%” but it doesn't appear in table 2.

- Some abbreviations should be specified: SWBS (line 356) and IT (line 370)

- I recommend homogenizing the typeface of the manuscript: lines 82-83, line 399, lines 407-410.
